# Effects of Fatigue Induced by Repeated Sprints on Sprint Biomechanics in Football Players: Should We Look at the Group or the Individual?

**DOI:** 10.3390/ijerph192214643

**Published:** 2022-11-08

**Authors:** Valentin Romero, Johan Lahti, Adrián Castaño Zambudio, Jurdan Mendiguchia, Pedro Jiménez Reyes, Jean-Benoît Morin

**Affiliations:** 1Centre for Sport Studies, Rey Juan Carlos University, 28942 Madrid, Spain; 2Laboratory of Human Motricity, Education Sport and Health (LAMHESS), Université Côte d’Azur, F-06200 Nice, France; 3Department of Physical Therapy, ZENTRUM Rehab and Performance Center, 31010 Barañain, Spain; 4Inter-University Laboratory of Human Movement Biology (LIBM EA 7424), University of Lyon, University Jean Monnet, F-42023 Saint Etienne, France

**Keywords:** fatigue, biomechanics, hamstrings, injuries, football

## Abstract

The aim of this study was to analyse the influence of fatigue on sprint biomechanics. Fifty-one football players performed twelve maximal 30 m sprints with 20 s recovery between each sprint. Sprint kinetics were computed from running speed data and a high-frequency camera (240 Hz) was used to study kinematic data. A cluster analysis (K-mean clustering) was conducted to classify individual kinematic adaptations. A large decrease in maximal power output and less efficiency in horizontally orienting the ground reaction force were observed in fatigued participants. In addition, individual changes in kinematic components were observed, and, according to the cluster analysis, five clusters were identified. Changes in trunk, knee, and hip angles led to an overall theoretical increase in hamstring strain for some players (Cluster 5, 20/51) but to an overall decrease for some others (Cluster 1, 11/51). This study showed that the repeated sprint ability (RSA) protocol had an impact on both kinetics and kinematics. Moreover, fatigue affected the kinematics in a different way for each player, and these individual changes were associated with either higher or lower hamstring length and thus strain.

## 1. Introduction

Football’s physical demands cover a wide spectrum of intensities during a match, from standing to walking, jogging, and sprinting [1,2]. However, only high-intensity actions have been related to decisive and crucial actions during matches [3,4]. During high-intensity running, the hamstrings display a peak in surface electromyography activity [5,6] but also estimated length and force peaks [7,8,9,10]. This maximal magnitude of hamstring activity (as assessed with surface electromyography) can only be observed through this type of running-based activity [11,12,13]. Over the last few years, a gradual increase in sprinting speed demands has been reported [14,15]. As a result, one of the major problems associated with this kind of effort (i.e., hamstring muscle injury) has increased [16,17,18,19,20,21,22].

Since up to 60% of reported injuries involving these muscles occur during high-speed actions [16], numerous studies have attempted to identify the sprint phase that presents the greatest risk for hamstring injury. Thus, some studies have determined the late swing phase to be the key phase for hamstring muscle injury [23,24,25,26], while others have established that it was the early stance [27,28,29]. However, due to their very short duration, these two phases can be considered as one, known as the “Swing–Stance Transition” [30]. Therefore, it seems logical to expect both phases of sprinting to be a key parameter in soccer both from a performance and injury prevention perspective [31]. Indeed, during late swing phase, it has been stated that the capability of the hamstring muscles to absorb the kinetic energy of the lower limb is key for sprint performance [32]. The ability to absorb this energy is particularly important during the early stance because strong opposing forces resulting from the ground reaction force are likely to increase the risk of hamstring injury [27,28,31]. Considering that many of the reported injuries occur while players perform under significant fatigue [16,22], it seems relevant to mention that recently higher high-speed running and lower recovery between high-speed bouts were reported immediately before an injury [33,34]. As a result, in fatigue conditions, muscles are injured at a similar length but fatigued muscles are likely able to absorb less energy before failure [35]. Consequently, this context is considered to be crucial for football-related injuries and may favour strain, which is the main cause of hamstring injury [10]. Strain has been defined as an excessive tensile load [36,37] which usually occurs when opposing tensile forces are simultaneously present [38], and during fatigue [35].

Assessing individual responses to the effects of fatigue on sprint kinetics and kinematics may help in understanding the complex reality of sprint-related hamstring injuries. Thus, when analysing the effects of a repeated sprints protocol, a significant decrease in performance can be observed together with a decrease in horizontal force production, not equally reflected in the total (i.e., resultant) ground reaction force recorded, which is practically unaltered by the fatigue protocols [5,39,40,41]. Similarly, the use of simple methodologies based on the position–time or speed–time data analysis is sensitive enough to assess the impact of this type of protocol on the ability to produce force. For example, studies by Nagahara et al. (2016) [42] or Jiménez-Reyes et al. (2019) [41] showed an unequal decrease in the theoretical maximal velocity and force outputs, the former being the most affected by fatigue.

However, the scientific literature has not only explored effects related to the kinetic component of sprint performance but also investigated the kinematics component. To our knowledge, only two studies have described the impact of acute fatigue on sprint kinematics and the associated risk of hamstring strain [43,44]. Only Small et al. (2009) [44] reported kinematic changes that could be associated with a greater hamstring strain, with an increase in anteroposterior pelvic tilt with fatigue during the entire stride. In contrast, Pinniger et al. (2000) [43] found that, when fatigued, athletes adopted a movement pattern that would be more protective (i.e., associated with less hamstring strain). Specifically, pilot studies showed that modifications in trunk, hip, or knee angle led to changes in hamstring length that were prospectively associated with hamstring injury risk [45,46,47,48,49].

Based on the available evidence, hamstring injury is a major issue in football, especially when players are fatigued. Thus, the ability to identify individuals showing running mechanics that can be associated with higher strain levels under ecological conditions could represent a major novelty in understanding the mechanisms of injury and more effectively targeting injury prevention strategies. Therefore, the objective of this study was to analyse the effect of fatigue on sprint kinetics and kinematics and the estimated hamstring strain resulting from a repeated sprint ability (RSA) protocol. We hypothesised that: (i) uniform adaptations would be reported for kinetic parameters, and (ii) non-uniform movement patterns would be evident between subjects during a fatigued condition.

## 2. Materials and Methods

### 2.1. Subjects

Fifty-one male football players (mean ± SD: age: 19.2 ± 1.74 years; height: 1.77 ± 0.06 m; body mass: 71.0 ± 9.79 kg) volunteered to participate in this study. All were sports science students and practiced in amateur competitive football. An informed consent was signed by all athletes on the day of testing, and the testing procedures were approved by the local review board and performed in the accredited research centre LAMHESS (#2016-08) in accordance with the Declaration of Helsinki.

### 2.2. Procedures

All participants performed the same warm-up. The warm-up included five minutes of low-intensity running, three dynamic stretch drills (hamstrings, gluteals, hip flexors), three sprint drills (knee lifts, A-skips, scissor runs), and three progressive accelerations (60, 80, and 100% of their maximal velocity). After warm-up, all participants performed a 30 m all-out sprint 5 min before the designed repeated sprint protocol. This protocol consisted of two sets of six 30 m all-out sprints with 20 s active recovery between each sprint and 3.5 min passive recovery between each set. All the players had already completed at least one RSA test with their respective clubs and were familiar with this type of effort. The test was performed on a synthetic pitch in the same weather conditions for all participants, and all protocols were performed in the morning. A verbal explanation of the protocol was given to each participant before the start of the session, and during the protocol verbal encouragement was provided.

### 2.3. Sprint Kinetics

For each sprint, sprint performance (split times 0–5, 0–10, and 0–20 m, in seconds) and mechanical outputs were computed using a radar device with a 46.875 Hz sampling frequency (Stalker ATS II Version 5.0.2.1, Applied Concepts, Dallas, TX, USA). Velocity–time data were collected using STATS software (Model: Stalker ATS II Version 5.0.2.1, Applied Concepts, Dallas, TX, USA) provided by the radar manufacturer. For each sprint, players accelerated from a standing position to their maximal velocity plateau. Individual linear sprint force–velocity (FV) profiles were then extrapolated from the fitting of the velocity–time data with an exponential function [50,51]. The method of Samozino et al. (2016) [50] was used to compute the sprint acceleration mechanical outputs based on an inverse dynamics approach applied to the centre of mass displacement. Thus, for each sprint the force/velocity profile was modelled [50], and individual force–velocity relationships were determined using the least squares regression method, with a time adjustment to ensure the actual start of the computation at t = 0 s, in case of delay between the time trigger and the actual increase in velocity [51]. Theoretical maximal horizontal force production (F0, N.kg^−1^) and theoretical maximal running velocity (V0, m.s^−1^) were calculated as the intercepts of the force–velocity relationship. Maximal mechanical power output associated with the horizontal component of the ground reaction force (Pmax, W.kg^−1^) was determined as F0 × V0/4. The ratio of force (RF) was computed as the ratio of the step-averaged horizontal component of the ground reaction force to the corresponding resultant force. Rate of decrease in RF (Drf), computed as the slope of the linear RF–velocity relationship, represented the decrement in RF with increasing speed [52]. The maximal value of the ratio of force (RFpeak, %) was identified as the highest RF value. Finally, for each sprint start, back steps (characterised by a quick step opposite the desired direction of travel [53]) were forbidden to prevent an alteration in performance during the first steps [53].

### 2.4. Sprint Kinematics

For all sprints, video images were obtained at 240 Hz using a smart phone video camera at an HD resolution of 720p (Iphone 6S, Apple Inc., Cupertino, CA, USA). The cameras were placed on tripods at 9 m perpendicular to the running lane and at the 25 m mark along a 0–30 m line, at hip height, allowing a 9 m field of view. A 5 m horizontal video calibration was recorded at each data collection session. The objective was to capture sagittal plane images during the maximal velocity phase from 22.5 m to 27.5 m, given that team sport athletes run at 95% to 100% of their maximal velocity within this section [54]. The same leg sequence was analysed before and after the RSA protocol for each player, with a specific focus of the experimenters to analyse the sequence as close to the midpoint of the camera as possible. The first two sprints (pre-fatigue condition) and the last two (fatigued condition) were analysed twice to improve reliability of the digital markers method. The kinematic sprint sequence of interest was the touchdown (first frame in which the foot was visibly in contact with the ground). The human body was modelled as 18 points (see Figure 1). The raw video files were imported into a digitising software (Kinovea, version 0.8.27) and manually digitised at full resolution with a zoom factor of x6. In order to facilitate the manual analysis, nine reflective markers were attached to each participant at specific locations (shoulder, elbow, and wrist joint centres, head of third metacarpal, greater trochanters, medial and lateral aspects of the knee joint centres, medial and lateral malleoli joint centres, and first distal metatarsal-phalangeal) to define body segments. The digitalised coordinates were exported to Excel (Microsoft Office 2019) where kinematics were determined. Angles of the trunk (relative to the horizontal), hips, and knees (ipsilateral and contralateral) were quantified. This study used the same procedures as in previous studies [55,56,57].

### 2.5. Statistical Analyses

All statistical analyses were conducted using Statistica software, version 7.1, except the cluster analysis which was conducted with JASP (version 0.14.1.0, University of Amsterdam, Netherlands). Before further statistical analyses, the normal distribution of the variables (Shapiro–Wilk test) and the homogeneity of the variances (Levene’s test) were confirmed (*p* > 0.05). A t-test was used to explore the effect of fatigue on kinetics and kinematic variables, and statistical significance was set at *p* < 0.05. All data are presented as average, standard deviation (SD), and coefficient of variation (CV). Additionally, Cohen’s d was computed for comparing effect sizes (ESs). ESs were classified as trivial (<0.2), small (0.2–0.49), moderate (0.5–0.79), and large (>0.8) (Cohen, 1988) with 95% confidence intervals (including lower limit and upper limit). Smallest worthwhile change was implemented (SWC = SD_pre and post pooled_ × 0.2) in order to determine meaningful change [58]. In addition, reliability statistical values were calculated, including intraclass correlation coefficient (ICC), coefficient of variation (CV), and minimum detectable change (MDC). Furthermore, a cluster analysis was conducted, and variables that have been reported in the literature to influence the hamstring strain level were selected, including trunk, ipsilateral knee and hip, and contralateral hip angles. According to the K-mean clustering and elbow method, the optimum number of clusters identified for analysis was 5. The model was optimised with respect to the Bayesian information criterion (BIC). Using the method of Khair et al. (2021) [59], clustering was repeated 5 times for each group and confirmed that each data point was consistently assigned to the same cluster group.

## 3. Results

### 3.1. Kinetics

The RSA protocol induced significant changes in kinetics for V0 (−12.9% ± 6.1; *p* < 0.01; ES: −1.99, large decrease), F0 (−6.98% ± 7.86; *p* < 0.01; ES: −0.65, moderate decrease), Pmax (−18.9% ± 8.41; *p* < 0.01; ES: −1.56, large decrease), RF peak (−6.42% ± 6.36; *p* < 0.01; ES: −0.94, large decrease), DRF (−9.64% ± 12.72; *p* < 0.01; ES: −0.71, moderate decrease), and time at 5, 10, and 20 m (6.33% ± 3.98; *p* < 0.01; ES: 1.23, large increase and 7.38% ± 3.86; *p* < 0.01; ES: 1.51, large increase and 9.00% ± 4.39; *p* < 0.01; ES: 1.80, large increase). All kinetic results are summarised in Table 1.

### 3.2. Kinematics

Fatigue induced a significant difference in touchdown kinematics with a more backward-inclined trunk angle (3.52% ± 5.44; *p* < 0.01; ES: 0.54, moderate increase); a more extended ipsilateral knee angle (1.84% ± 4.18; *p* = 0.003; ES: 0.43, small increase); and a more extended contralateral hip angle (5.98% ± 6.46; *p* < 0.01; ES: 1.16, large increase). Subjects who significantly increased their values with fatigue showed a significant difference with a more backward-inclined trunk angle (5.87 % ± 3.69; *p* < 0.01; ES: 1.03, large increase); a more extended ipsilateral knee angle (4.18% ± 3.68; *p* < 0.01; ES: 0.93 large increase); and a more extended ipsilateral and contralateral hip angle (5.87% ± 2.98; *p* < 0.01; ES: 1.31, large increase and 8.46% ± 4.91; *p* < 0.01; ES: 1.95, large increase). Additionally, some subjects showed opposite changes with fatigue, and they showed significant changes with a more forward-inclined trunk angle (−5.25% ± 5.79; *p* = 0.072; ES: −0.89, large decrease) and a less extended ipsilateral knee and hip angle (−2.24% ± 2.14; *p* = 0.003; ES: −0.85, large decrease and −7.07% ± 5.80; *p* < 0.01; ES: −1.70, large decrease) and contralateral hip angle (−4.98 ± 4.39; *p* = 0.067; ES: −1.03, large decrease).

Cluster analysis showed that a five-cluster solution provided optimal treatment of variance. Thus, it was decided to use five cluster groups to minimise within-group variance (increase homogeneity) and maximise between-group variance. In addition, the R^2^ value of the K-mean clustering was 0.75. The standard score is summarised in Table 2, and the density change is reported in Figure 2. The characteristics of each cluster were: Cluster 1 (C1) (n = 11, within-cluster heterogeneity = 0.21 and silhouette score = 0.40), Cluster 2 (C2) (n = 16, 0.23 and 0.39), Cluster 3 (C3) (n = 1, 0.00 and 0.00), Cluster 4 (C4) (n = 3, 0.06 and 0.58), and Cluster 5 (C5) (n = 20, 0.50 and 0.19). The within sum of squares for each subgroup was: C1 = 10.94, C2 = 11.81, C3 = 0.00, C4 = 3.13, and C5 = 26.08. All the sprint kinematic variables are visualised in Figure 3.

## 4. Discussion

The purpose of this study was to investigate the effects of a repeated sprint protocol on sprint kinetics and kinematics in amateur football players. The main findings show that (i) fatigue affects both sprint kinetics and kinematics, and (ii) contrasted individual changes in maximal velocity sprint pattern justify an individual analysis to assess the effects of fatigue on sprint kinematics.

Regarding kinetic components, similar findings were obtained compared to previous studies. A homogeneous response can be observed across the main determinants of the mechanical characteristics (50/51, 36/51, and 50/51 players presented significant decreases in V0, F0, and Pmax, respectively), suggesting that the fatigue induced by repeated sprints led to similar overall responses for the large majority of individuals. Furthermore, the magnitude of the responses discussed in this intervention support the idea that the decrease in Pmax seems to be strongly affected during RSA-based fatigue protocols and more related to an impairment in force production at maximal velocity (V0, large) than at low velocity (early acceleration F0, moderate).

In contrast, kinematic responses to fatigue appear to be more individual, and two players with similar changes in kinetics may show contrasted kinematic changes. This fact can be clearly exemplified by analysing the effects of the protocol on hip angle: when analysing the response induced by fatigue on this joint in a traditional way, the individual distribution of these responses (18/20/13) is masked behind the mean values (pre: 135.2 ± 13.8 vs. post: 135.2 ± 12.5 degrees). While approaches such as SWC allow us to identify that when the change observed in an athlete is individually relevant, they are still ineffective tools for identifying responses to such efforts as they do not consider how the involved joint positions interact and consequently how these affect hamstring length and thus strain. Therefore, the main novelty of this study was that it considered the individual response to the protocol in a non-isolated way, which enabled the detection of different “patterns” that may occur when fatigue becomes apparent. As suggested by Morin et al. (2020) [60] and Welch et al. (2020) [61], details may be lost with between-groups analysis when compared to individual analysis. According to a preliminary study by Sprague and Mann (1983) [62], adaptations to fatigue are individual and depend on different factors. The current study reported individual characteristics in kinematic adaptations between different sprinters when fatigued, suggesting that individual strategies are applied when an athlete is exposed to repeated sprints. Accordingly, while some athletes will adopt a “protective” movement pattern (i.e., likely associated with reduced hamstring length and thus strain (C1, 11/51)), others adopt an overall pattern that potentially increases hamstring length, strain, and thus, all other things equal, risk of injury (C5, 20/51). Thus, while the existing evidence has theorised contradictory responses to repeated sprint fatigue protocols (increased risk of injury versus more “protective” mechanisms), this new approach sheds some light on the limited evidence by scientifically supporting both scenarios under a single focus.

Based on the cluster analysis, three kinds of patterns can be differentiated: clear theoretical increases or decreases in injury risk, and non-uniform patterns. According to Higashihara et al. (2015) [47], greater trunk forward tilt is associated with greater anterior pelvic tilt, and consequently increased hamstring strain during sprinting [44]. Assuming strain as the major determinant of tissue failure, a greater anterior pelvic tilt would superiorly translate the ischial tuberosity resulting in a greater active lengthening and passive tension demand of the posterior thigh musculature. Moreover, during sprinting, hamstring injury primarily involves the proximal region compared to the distal region of the long head of the biceps femoris [63]. This result, based on the pelvic action, is the main source of strain on the hamstrings due to the large lever arm. The aforementioned arguments may explain the association found between anterior pelvic tilt and hamstring strain injury risk in different prospective studies and in C5 (n = 20) during fatigue. Inversely, it seems logical to expect that, anatomically, a more extended trunk together with a more posterior pelvic tilt would reduce the tensile strength imposed on the hamstrings during high-speed running as happened in C1 (n = 11). Additionally, the accumulation of metabolites (H+ and Pi ions) during RSA exercises has been shown to potentially alter muscle activity and reduce the capacity to generate force [43,64]. This may alter the athlete’s overall kinetic energy absorption capacity [65], compromising eccentric capacity prior to the landing phase [66], where the hamstring muscles act to decelerate the forward leg motion [67]. Thus, it seems important to consider that fatigued muscles are likely able to absorb less energy before failure [35]. Based on these results, it is suggested that fatigue impairs the ability of the hamstring muscles to limit knee extension. Consequently, when an increase in knee extension occurs simultaneously with an increase in ipsilateral hip flexion, a phenomenon known in the literature as over-stride is observed [68,69]. The arrangement displayed in the front leg, combined with a greater distance to the centre of mass at the moment of landing, implies greater tensile force indexes on the hamstrings, potentially increasing the risk of injury [28], as observed in our cohort study with C5. Additionally, an increase in contralateral hip extension, as reported for C5, affects hamstring length [70], particularly via the action of the contralateral iliopsoas on the pelvis [46,71]. To summarise, C1 (n = 11) presented articular changes that considerably reduced strain level, and, inversely, C5 (n = 20), clearly increased the initial strain level during fatigue.

Finally, C2, C3, and C4, which included 20 subjects, presented non-uniform strategies. In other words, C2 and C3 showed an increase in trunk forward inclination and an increase in ipsilateral hip flexion. These changes were described as increasing hamstring length and thus potentially strain levels. However, at the same time, these groups decreased ipsilateral knee extension and contralateral hip extension. C4 showed a decrease in trunk tilt, which implies a decrease in strain level but increases in ipsilateral hip flexion, ipsilateral knee extension, and contralateral hip extension. These changes can therefore “offset” each other as some joints have a risky movement pattern, which is counter-balanced by other joints.

To our knowledge, only two studies have investigated fatigue impacts on sprint kinematics and risk of hamstring strain injury, but these showed opposite results. Pinniger et al. (2000) [43] found that in fatigue conditions mechanisms were implemented that induced a protective movement pattern against hamstring injury. In contrast, Small et al. (2009) [44] reported movement patterns that increased the risk of hamstring injury. These results clearly show the importance of the individual characteristics of each person in their responses to fatigue. Indeed, two other studies have also described the impact of fatigue on the knee joint [72,73]. The two studies mentioned above showed opposite results. In fact, Wilmes et al. (2021) [72] reported an increase in peak knee extension, while Baumert et al. (2021) [73] showed a decrease in peak knee extension during fatigue. In the present study, some athletes adopted a movement pattern that would protect them from hamstring strain, but others showed body arrangements that potentially increased this strain and in turn probably their risk of hamstring injuries. Interestingly, depending on the player, the change (protective or at-risk) affects different joints. The possible explanatory mechanisms for these changes support the importance of analysing individual adaptations to fatigue when managing athletes in the context of hamstring injury prevention.

This study potentially provides valuable practical information for team sport coaches and strength and conditioning coaches. The results demonstrate that (i) the RSA protocol induces changes in both sprint kinetics and kinematics; (ii) changes in kinematics show highly variable patterns between players; and (iii) individual changes in kinematics may increase or decrease the theoretical risk of hamstring injury. These observations can help coaches better orient and individualise both RSA training and hamstring injury prevention.

The main limitation of this study relates to the kinematic analysis, since the video recording system only provided 2D results obtained in the sagittal plane. However, our method has been previously used with high reliability [55,56,57]. Additionally, although it appears to play an essential role in hamstring strain, specific pelvic orientation cannot be measured using this kind of analysis, which limits the discussion to information inferred from trunk position. Nevertheless, some authors have correlated the movement of the trunk and pelvis [47], and our present study measures the trunk angulation. Finally, it has been shown that fatigue can increase or decrease the level of hamstring strain, without this reflecting a high or low likelihood of injury. Factors such as starting position can play an essential role and should be considered in establishing these likelihood thresholds. However, such analysis was not performed in this study because of the large sample size required for an accurate model.

## 5. Conclusions

Fatigue induced by a repeated sprints protocol affected not only sprinting kinetics but also kinematics. The decrease in power output found in our study mainly results from a decrease in the maximal velocity component. The experimental results obtained in the present study also showed that fatigue affects the kinematics in a different way for each player. Indeed, subjects can have patterns of movement that could be considered as either protective (i.e., inducing lower hamstring length change) or riskier in relation to hamstring injury. Furthermore, it seems relevant to test the kinetics and kinematics during an RSA test in order to better individualise both sprint training and hamstring injury prevention.

## Figures and Tables

**Figure 1 ijerph-19-14643-f001:**
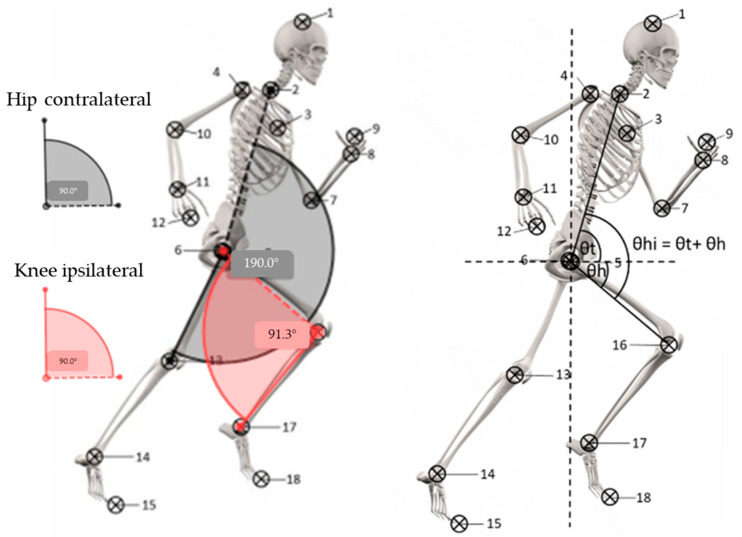
Human body modelling (18 points method: vertex of the head, halfway between the suprasternal notch and the 7th cervical vertebra, shoulder, elbow, and wrist joint centres, head of third metacarpal, hip, knee, and ankle joint centres, and the tip of the toe) and angle calculation methods.

**Figure 2 ijerph-19-14643-f002:**
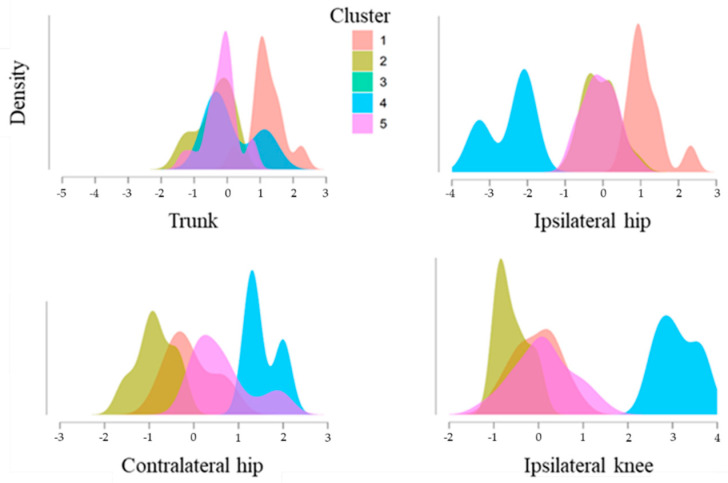
Density plots of cluster change according to the standard score.

**Figure 3 ijerph-19-14643-f003:**
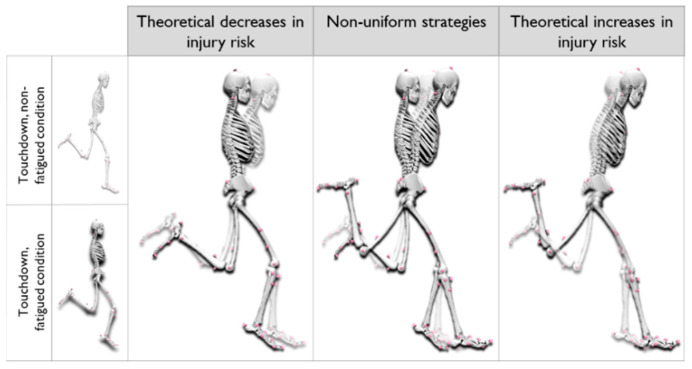
Schematic representation of sprint kinematic changes during touchdown for the most influencing segments (trunk, knee, and hip angles) on hamstring strain injury. Comparison between non-fatigued and fatigued conditions.

**Table 1 ijerph-19-14643-t001:** Changes in kinetic variables.

Variable	Pre x ± SD (CV)	Post x ± SD (CV)	%∆ ± SD	*p*-Value	Effect Size (Lower/Upper 95% CI)	Descriptor	Individual Response (Increase/No Change/Decrease)
Vmax theoretical V0 (m/s)	8.53 ± 0.47 (5.53)	7.43 ± 0.62 (8.39)	−12.89± 6.09	<0.01	−1.99 (−2.06/–1.92)	Large decrease	0-1-50
Fmax theoretical F0 (N/kg)	7.50 ± 0.76 (10.08)	6.98 ± 0.85 (12.18)	−6.98 ± 7.86	<0.01	−0.65 (−0.71/–0.60)	Moderate decrease	6-9-36
Pmax (W/kg)	15.86 ± 1.88 (11.84)	12.86 ± 1.95 (15.19)	−18.90 ± 8.41	<0.01	−1.56 (−1.63/–1.50)	Large decrease	0-1-50
RFpeak	0.50 ± 0.03 (6.18)	0.47 ± 0.04 (7.89)	−6.42 ± 6.36	<0.01	−0.94 (−1.00/–0.89)	Large decrease	5-6-40
DRF	−0.08 ± 0.01 (10.62)	−0.09 ± 0.01 (−14.46)	−9.64 ± 12.72	<0.01	−0.71 (−0.77/–0.66)	Moderate decrease	11-4-36
Time @ 5 m (s)	1.39 ± 0.06 (4.60)	1.48 ± 0.08 (5.30)	6.33 ± 3.98	<0.01	1.23 (1.17/1.29)	Large increase	47-4-0
Time @ 10 m (s)	2.15 ± 0.09 (4.24)	2.31 ± 0.12 (5.10)	7.38 ± 3.86	<0.01	1.51 (1.45/1.57)	Large increase	50-1-0
Time @ 20 m (s)	3.48 ± 0.14 (4.12)	3.80 ± 0.20 (5.29)	9.00 ± 4.39	<0.01	1.80 (1.73/1.86)	Large increase	51-0-0

**Table 2 ijerph-19-14643-t002:** Changes in kinematic variables and standard score according to cluster analysis during touchdown phase.

Touchdown Parameters				
Group	Trunk Angle	Hip Angle	Knee Angle	Contralateral Hip Angle
Pre x ± SD (CV)	75.22 ± 4.81 (6.39)	135.18 ± 13.83 (5.47)	149.07 ± 6.63 (4.45)	170.24 ± 9.03 (5.30)
Post x ± SD (CV)	77.74 ± 4.47 (5.76)	135.23 ± 12.51 (4.51)	151.63 ± 5.30 (3.50)	180.02 ± 7.82 (4.35)
%∆ ± SD	3.52 ± 5.44	0.28 ± 6.09	1.84 ± 4.18	5.98 ± 6.46
*p*-value	<0.001	0.964	0.003	<0.001
Effect Size (Lower/Upper 95% CI)	0.54 (0.49/0.60)	0.01 (−0.05/0.06)	0.43 (0.37/0.48)	1.16 (1.10/1.22)
Descriptor	Moderate increase	Trivial	Small increase	Large increase
Individual Response (Increase/No change/Decrease)	36-8-7	18-20-13	30-7-14	39-7-5
Standard Score				
Cluster	Trunk Angle	Hip Angle	Knee Angle	Contralateral Hip Angle
C1 (n = 11)	1.231	1.151	−0.053	−0.069
C2 (n = 16)	−0.393	−0.048	−0.565	−0.869
C3 (n = 1)	−3.989	−2.638	−2.658	−2.864
C4 (n = 3)	0.160	−2.464	2.823	1.537
C5 (n = 20)	−0.187	−0.093	0.132	0.646
RELIABILITY (SPRINT 1 vs. 2)				
	Trunk Angle	Hip Angle	Knee Angle	Contralateral Hip Angle
ICC (Lower/Upper 95% CI)	0.97 (0.95/0.98)	0.99 (0.98/0.99)	0.98 (0.97/0.99)	0.99 (0.98/1.00)
CV % (Lower/Upper 95% CI)	0.65 (0.25/1.05)	0.46 (0.30/0.62)	0.47 (0.28/0.67)	0.37 (0.21/0.52)
MDC	2.32	2.16	2.66	2.38
MDC (%)	3.09	1.60	1.78	1.40

## Data Availability

The data presented in this study are available on request from the corresponding author.

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
