# Peer review of "Effects of Fatigue Induced by Repeated Sprints on Sprint Biomechanics in Football Players: Should We Look at the Group or the Individual?"

_ijerph, 2022, doi:10.3390/ijerph192214643_

Round 1

Reviewer 1 Report

Comments to the Author

Thank you for your manuscript entitled ‘Effects of fatigue induced by repeated sprints on sprint biomechanics in football players: should we look at the group or the individual?’. This manuscript brings an interesting topic for practitioners, strength and conditioning coaches, and researchers, exploring the effect of fatigue on sprinting and consequently on the increased risk of hamstring muscle injuries, considering also individual adaptations. This is an original work with an interesting approach. Indeed, trying to individualize this analysis is a centrepiece in this whole process. The literature is full of studies focused on the averages of a given group under study, which hides some important traces of reality. Each subject is an individual and presents specific individualized adaptations depending on several factors. I further believe that the study could be widely cited in the future. The abstract and the methods are clearly explained as well as the results and the discussion builds on and contextualises the results and is interesting. However, I have some minor comments regarding the presentation and interpretation of these results but nothing that you should not be able to address relatively quickly.

1-     The authors used a 2D analysis instead of a 3D one. Although this approach has limitations, the most important changes occur in the sagittal plane which gives an idea of the changes in the length of these muscles. Please justify the good reliability of 2D by previous studies.

2-     Please described why only one discrete time instance during the sprint via digitization of video footage was analysed.  

3-     We would like to draw the attention of the authors to figure 1 since it is not clear what the pre and the post are in the figure. Please review the captions and group the figures in pairs, matching the figure placed in the left column with the same in the right column, which facilitates the analysis.

Other questions

1-     Please describe the warm-up. Was it standardised?

2-     Can the authors describe how fatigue was determined? Or was it assumed? Also, were the participants activities controlled prior to testing? How did you control if the participants were fatigued prior to participating or not?

3-     Did you estimate the hamstrings tension, how?

4-     What do these numbers (18/20/13), mean?

5-     Can you describe future researches required? 

Author Response

Point 1: Thank you for your manuscript entitled ‘Effects of fatigue induced by repeated sprints on sprint biomechanics in football players: should we look at the group or the individual?’. This manuscript brings an interesting topic for practitioners, strength and conditioning coaches, and researchers, exploring the effect of fatigue on sprinting and consequently on the increased risk of hamstring muscle injuries, considering also individual adaptations. This is an original work with an interesting approach. Indeed, trying to individualize this analysis is a centrepiece in this whole process. The literature is full of studies focused on the averages of a given group under study, which hides some important traces of reality. Each subject is an individual and presents specific individualized adaptations depending on several factors. I further believe that the study could be widely cited in the future. The abstract and the methods are clearly explained as well as the results and the discussion builds on and contextualises the results and is interesting. However, I have some minor comments regarding the presentation and interpretation of these results but nothing that you should not be able to address relatively quickly.

Response 1: We thank the reviewer for the positive and constructive comments on our paper. We did our best to follow the recommendations made and we hope this will overall improve the quality of our work.

Point 2: The authors used a 2D analysis instead of a 3D one. Although this approach has limitations, the most important changes occur in the sagittal plane which gives an idea of the changes in the length of these muscles. Please justify the good reliability of 2D by previous studies.

Response 2: Reliability in our study was good (see Table. 2). Our method has been previously used with high reliability by Wild et al. (2018). Therefore, we think that this limitation is acceptable as long as it is discussed. In addition, the limitation of using a 2D field method to quantify running mechanics is outweighed by the possibility to assess sprint fatigue in a very applied field context, which was not possible in laboratory conditions.

Point 3: Please described why only one discrete time instance during the sprint via digitization of video footage was analysed.  

Response 3: In the literature, there is some debate as to which phase of sprinting is most at risk for hamstring injury. Indeed, some authors have described the "late swing" as the most critical phase for the hamstrings (Chumanov et al., 2011; Chumanov et al., 2012; Schache et al., 2012; Yu et al., 2008) while others have identified the "touchdown" (Mann et al., 1980; Orchard et al., 2012). However, some authors describe these two phases as one: "the swing-stance transition" (Liu et al., 2017). In addition, the touchdown phase is easier to identify. For these reasons, a discrete analysis was made and the touchdown was chosen. It should be noticed that these two phases occur so close in time that to date no technology allows clearly solving the debate. From a mechanical strain perspective, these two phases are probably very close to we think our choice makes sense regardless.

Point 4: We would like to draw the attention of the authors to figure 1 since it is not clear what the pre and the post are in the figure. Please review the captions and group the figures in pairs, matching the figure placed in the left column with the same in the right column, which facilitates the analysis.

Response 4: Here, the figure 1 does not present PRE/POST changes but focuses on the density of each cluster group. The subjects were grouped in this way (by a cluster analysis) according to their personal adaptation to the fatigue condition.

Point 5: Please describe the warm-up. Was it standardised?

Response 5: Thanks, we have now added a description of warm-up. All participants performed the same warm-up, which included five minutes low-intensity running, three dynamic stretch drills (hamstrings, gluteals, hip flexors), three sprint drills (knee lifts, A-skips, scissor runs), and three pro­gressive accelerations (60, 80 and 100% of their maximal velocity).

Point 6: Can the authors describe how fatigue was determined? Or was it assumed? Also, were the participants activities controlled prior to testing? How did you control if the participants were fatigued prior to participating or not?

Response 6: Fatigue was assumed as a decrease in sprint performance output, according to the effect produced by an RSA protocol and described in a previous study (Girard et al. 2011). In addition, fatigue was characterized by a decrease in sprint performance in all subjects (i.e. beyond the smallest worthwhile change computed). For example, the RSA protocol induced significant changes for V0 (12.9% ± 6.1 ; p<0.01; ES : -1.99, large decrease; SWC: 0-1-50) and time at 20-m (9.00% ± 4,39 ; p<0.01 ; ES : 1.80, large increase; SWC: 51-0-0). The subjects were asked not to train in the 48 hours before the study so that they would arrive in a good state of physical condition. A verbal agreement was therefore made with them.

Point 7: Did you estimate the hamstrings tension, how?

Response 7: Hamstrings tension was estimated according to anatomy and biomechanics pattern movements (Hawkins et Hull., 1990; Small et al., 2009; Souza., 2016).

Point 8: What do these numbers (18/20/13), mean?

Response 8: These numbers are calculated according to the Smallest Worthwhile Change in order to determine meaningful change. This provides an indication of the individual sprint pattern adaptation for each subject (PRE/POST comparison). Indeed, our study includes 51 subjects and, in this example, (contralateral hip), 18 subjects had a significative decrease of hip flexion; 20 have not changed significantly (i.e. beyond the smallest worthwhile change computed) and 13 showed a significative increase of hip flexion. This information corresponds to the individual response (Increase/No change/De-crease) in table 1 and 2.

Point 9: Can you describe future researches required?

Response 9: Future research should focus on the trainability of these sprint patterns in fatigue condition. A recent study showed that it was possible to modify the biomechanics of sprinting (Mendiguchia et al., 2021), but is there any training that can modify the sprint pattern and maintain it under fatigue conditions?

Reviewer 2 Report

The manuscript entitled “Effects of fatigue induced by repeated sprints on sprint biomechanics in football players: should we look at the group or the individual? “was reviewed as stated further. This study investigated the effects of fatigue on sprint biomechanics through analyzing the sprint kinematic data in 51 footballer players using cluster analysis. They demonstrated that fatigue affected the kinetics (and kinematics) differently for each player and these individual changes were associated with either higher or lower hamstring length or strain. This manuscript require some minor changes to be suitable for publication in the Journal. My comments on this manuscript are formulated as follows:

1.       Introduction

This part has satisfactorily been written. In my opinion the need to perform this study was sufficiently explained.

2.       Materials and methods

In general, the method section has been written with all required details. However, the below changes should be applied.

Section 2.4. Sprint Kinematics

Line 146: In my opinion, it would be easier to follow if the authors provide a picture to “18 points” for human body modelling.

3.       Discussion

Line 256: can the authors name some of the factors influencing the adaptation to fatigue? The authors have made it clear according to their work and references 43 and 44 that protective movement pattern is individual, but which factors they have considered. Have they considered any differences between players which led to specific behavioral pattern towards fatigue, e.g the players’ body mass index, or their professional years of playing etc. if yes, please comment on this in the manuscript.

Is there any research doing similar study on female football players to compare with your results and if women take different protective approach toward fatigue.

Line 315: an extra space btw. “in” and “peak”

Author Response

Point 1: The manuscript entitled “Effects of fatigue induced by repeated sprints on sprint biomechanics in football players: should we look at the group or the individual? “was reviewed as stated further. This study investigated the effects of fatigue on sprint biomechanics through analyzing the sprint kinematic data in 51 footballer players using cluster analysis. They demonstrated that fatigue affected the kinetics (and kinematics) differently for each player and these individual changes were associated with either higher or lower hamstring length or strain. This manuscript require some minor changes to be suitable for publication in the Journal. My comments on this manuscript are formulated as follows:

Response: We thank the reviewer for the positive and constructive comments on our paper. We did our best to follow the recommendations made and we hope this will overall improve the quality of our work.

  1. Introduction

Point 2: This part has satisfactorily been written. In my opinion the need to perform this study was sufficiently explained.

Response: We thank the reviewer for his comment.

  1. Materials and methods

Point 3: In general, the method section has been written with all required details. However, the below changes should be applied.

Section 2.4. Sprint Kinematics

Line 146: In my opinion, it would be easier to follow if the authors provide a picture to “18 points” for human body modelling.

Response: Thanks, we have now added a figure in order to represent the human body modelling and the angles calculation (Figure. 1).

  1. Discussion

Point 4: Line 256: can the authors name some of the factors influencing the adaptation to fatigue? The authors have made it clear according to their work and references 43 and 44 that protective movement pattern is individual, but which factors they have considered. Have they considered any differences between players which led to specific behavioral pattern towards fatigue, e.g the players’ body mass index, or their professional years of playing etc. if yes, please comment on this in the manuscript.

Response: We believe that the adaptations during fatigue can vary a lot between individuals yes, but in the present study no correlation was made to explain the reason of these adaptations. Indeed, BMI or level of play were not used to create any correlation tests. However, one of the reasons to consider would probably be the initial sprint pattern. Nevertheless, the key message remains that there are indeed adaptations to fatigue and that they are individual (i.e. not systematically observed for all individuals).

Point 5: Is there any research doing similar study on female football players to compare with your results and if women take different protective approach toward fatigue.

Response: To our knowledge no study investigated the impact of an RSA protocol on sprinting biomechanics on female football players. One recent study reported individuals’ adaptation during fatigue condition in elite female football players (Zago et al., 2021). Nevertheless, this study reported the impact on lower limb biomechanics and implications for ACL injury prevention. The aim of this study was to investigate COD strategies during fatigue but not during linear maximal sprinting.

Point 6: Line 315: an extra space btw. “in” and “peak”

Response: Corrected, thank you.

Reviewer 3 Report

Thank you for the possibility to review the present work and congratulations to the authors for carrying out the present study.

The main aim is clearly described as the authors analysed the influence of fatigue on sprint biomechanics in football players. However, the main concern of the reviewer is that the authors did not define clearly fatigue and how they measured/controlled fatigue during the experiment (which marker has been used and referred to which component of fatigue?). A clarification of this aspect may help the reader to take away the main findings and potentially to apply them to other sport or exercise contextes than football.

Here below are my general and specific comments.

General:

Introduction section could be cleared and probably English editing could help. The information flow is clear and the hypotheses clearly stated, however, the text does not flow very well when reading, especially in the first part.

Methods section:

It was enough to acquire quality images along the 25m sprint a single "camera" (a smartphone)? 

Main concern: how the authors established a fatigued and non fatigued condition? and what kind of fatigue the authors refer to?

An illustration in the methods section representing the study protocols and a description of the variables analysed would help the reader.

Discussion and conclusions

Same comments as for the introduction section.

Specific: 

Abstract line 24: please define RSA for the reader.

Introduction line 58: maybe better tensile load rather than "tensile strength"

Author Response

Point 1: Thank you for the possibility to review the present work and congratulations to the authors for carrying out the present study. The main aim is clearly described as the authors analysed the influence of fatigue on sprint biomechanics in football players. However, the main concern of the reviewer is that the authors did not define clearly fatigue and how they measured/controlled fatigue during the experiment (which marker has been used and referred to which component of fatigue?). A clarification of this aspect may help the reader to take away the main findings and potentially to apply them to other sport or exercise contextes than football.

Response 1: We thank the reviewer for this positive and constructive comment on our paper. We did our best to follow the recommendations made and we hope this will overall improve the quality of our work. Fatigue was assumed according to the effect produced by an RSA protocol and described in a previous study (Girard et al. 2011). In addition, fatigue was characterized by a decrease in sprint performance in all subjects (i.e. beyond the smallest worthwhile change computed).

Here below are my general and specific comments.

General:

Point 2: Introduction section could be cleared and probably English editing could help. The information flow is clear and the hypotheses clearly stated, however, the text does not flow very well when reading, especially in the first part.

Response 2: Thank you. We have now had an external source check the language.

Methods section:

Point 3: It was enough to acquire quality images along the 25m sprint a single "camera" (a smartphone)? 

Response 3: Only the max velocity phase was analyzed. That means kinematic was registered between 22.5 and 27.5 meters. We filmed with a smartphone but we believe that a 240-fps camera within such a volume is a very good standard in locomotion biomechanics. Most importantly our reliability analysis showed good results.

Point 4: Main concern: how the authors established a fatigued and non fatigued condition? and what kind of fatigue the authors refer to?

Response 4: After the warm-up, each subject completed a free sprint and then recovered for a period of 3.5 minutes. Then the first sprint of the RSA test was performed. The fatigue condition includes the last two sprints (11 and 12) of the RSA protocol. Fatigue was characterized by a decrease in sprint performance in all subjects (i.e. beyond the smallest worthwhile change computed). For example, the RSA protocol induced significant changes for V0 (-12.9% ± 6.1 ; p<0.01; ES : -1.99, large decrease; SWC: 0-1-50) and time at 20-m (9.00% ± 4,39 ; p<0.01 ; ES : 1.80, large increase; SWC: 51-0-0).

Point 5: An illustration in the methods section representing the study protocols and a description of the variables analysed would help the reader.

Response 5: According to this comment and another reviewer’s comment, we have now added a figure (see Figure. 1) in order to represent the angles calculation and the human body modelling.

Discussion and conclusions

Point 6: Same comments as for the introduction section.

Response 6: Thank you. We have now had an external source check the language within the manuscript.

Specific: 

Point 7: Abstract line 24: please define RSA for the reader.

Response 7: Defined, thank you.

Point 8: Introduction line 58: maybe better tensile load rather than "tensile strength"

Response 8: Corrected, thank you.

Reviewer 4 Report

This study is really interesting,as the aim of this study was to analyse the influence of fatigue on sprint biomechanics. As a lot of information is present, authors used a lot of abbreviations. They should think of using it less, as some parts of the paper is hard to follow.

Why did you use iphone 6s if you did this research in last few years? There are much better cameras on modern phones.

Please add country of origin for this phone

Author Response

Point 1: This study is really interesting,as the aim of this study was to analyse the influence of fatigue on sprint biomechanics. As a lot of information is present, authors used a lot of abbreviations. They should think of using it less, as some parts of the paper is hard to follow.

Response 1: We thank the reviewer for this comment and helpful recommendations. We hope our responses and revisions will overall improve the content of this work.

Point 2: Why did you use iphone 6s if you did this research in last few years? There are much better cameras on modern phones.

Response 2: We have taken into consideration the camera used but assuming the resolution of the smartphone, we believe that a 240-fps camera is a good standard in locomotion biomechanics. In addition, since the iPhone 6 model, the slow-motion frame rate has not increased over time (most modern Apple devices still include the 240 fps and 1080 ppi resolution). So basically, most recent iPhones or iPads provide the same slow motion video quality than previous devices up to the iPhone 6. More importantly, the reliability showed to be good with the iphone 6 in this study and our pilot studies.

Point 3: Please add country of origin for this phone

Response 3: Corrected, thank you.

Reviewer 5 Report

Dear Authors,

I congratulate you for your thorough and detailed study on the effect of fatigue on kinetics, kinematics of sprinting that could help the individual understanding of hamstring strain and in turn injury risk.

My only concern is the lack of the indication on the familiarisation protocol underwent by the players before starting the experimental sessions.

Around line 104 you should specify if and how the subjects were familiarised with the experimental protocol.

Author Response

Point 1: I congratulate you for your thorough and detailed study on the effect of fatigue on kinetics, kinematics of sprinting that could help the individual understanding of hamstring strain and in turn injury risk.

Response 1: We thank the reviewer for his comment.

Point 2: My only concern is the lack of the indication on the familiarisation protocol underwent by the players before starting the experimental sessions.

Response 2: All the players had already completed an RSA test with their respective clubs an were fully familiar with this type of effort and task.

Point 3: Around line 104 you should specify if and how the subjects were familiarised with the experimental protocol.

Response 3: We have now added this, apologies for forgetting.

Round 2

Reviewer 3 Report

Thanks to the authors for accepting my suggestions.

The work improved consistently and all issues are resolved.

The manuscript reads now more fluently and both, methodological aspects and discussion of the results are more clear to the reader.

Good luck for next steps of the revision process, I don't have further comments.